Strong geographical variation in wing aspect ratio of a damselfly, Calopteryx maculata (Odonata: Zygoptera)

Hassall Christopher c.hassall@leeds.ac.uk
School of Biology, University of Leeds , Leeds , United Kingdom
Brady Sean
Electronic publication date: 2015 Aug 25
Publication date: 2015
Volume: 3
Electronic Location ID: e1219
Received 2015 Apr 21; Accepted 2015 Aug 7
Copyright: © 2015 Hassall
Copyright year: 2015
Copyright holder: Hassall
License: This is an open access article distributed under the terms of the Creative Commons Attribution License, which permits unrestricted use, distribution, reproduction and adaptation in any medium and for any purpose provided that it is properly attributed. For attribution, the original author(s), title, publication source (PeerJ) and either DOI or URL of the article must be cited.
License URL: https://creativecommons.org/licenses/by/4.0/

Keywords: Wing morphology, Aspect ratio, Dispersal, Flight, Damselfly, Range, Odonata

Funding: British Ecological Society Small Ecological Project Grant Government of Canada Postdoctoral Fellowship Ontario MRI Fellowship The study was funded by a British Ecological Society Small Ecological Project Grant and I was supported by a Government of Canada Postdoctoral Fellowship and an Ontario MRI Fellowship. The funders had no role in study design, data collection and analysis, decision to publish, or preparation of the manuscript.

==============================
Geographical patterns in body size have been described across a wide range of species, leading to the development of a series of fundamental biological rules. However, shape variables are less well-described despite having substantial consequences for organism performance. Wing aspect ratio (AR) has been proposed as a key shape parameter that determines function in flying animals, with high AR corresponding to longer, thinner wings that promote high manoeuvrability, low speed flight, and low AR corresponding to shorter, broader wings that promote high efficiency long distance flight. From this principle it might be predicted that populations living in cooler areas would exhibit low AR wings to compensate for reduced muscle efficiency at lower temperatures. I test this hypothesis using the riverine damselfly, Calopteryx maculata, sampled from 34 sites across its range margin in North America. Nine hundred and seven male specimens were captured from across the 34 sites (mean = 26.7 ± 2.9 SE per site), dissected and measured to quantify the area and length of all four wings. Geometric morphometrics were employed to investigate geographical variation in wing shape. The majority of variation in wing shape involved changes in wing aspect ratio, confirmed independently by geometric morphometrics and wing measurements. There was a strong negative relationship between wing aspect ratio and the maximum temperature of the warmest month which varies from west-east in North America, creating a positive relationship with longitude. This pattern suggests that higher aspect ratio may be associated with areas in which greater flight efficiency is required: regions of lower temperatures during the flight season. I discuss my findings in light of research of the functional ecology of wing shape across vertebrate and invertebrate taxa.

Introduction

Powered flight has evolved independently in four different lineages: the pterosaurs, insects, birds, and bats, allowing animals to exploit novel niches and avoid predators. The adaptations that allowed each of these transitions to an aerial niche represent a suite of similar traits that can be broken down into a number of functional morphological components that influence inter- and intraspecific variation in flight performance. First, absolute body size is correlated with dispersal ability across a wide range of taxa (Jenkins et al., 2007). Second, the ratio of body mass to wing area—known as “wing loading”—has a strong influence on the amount of thrust generated per wingbeat (Dudley, 2002). However, for the purposes of this study I am most interested in the third component of variation: that of wing shape. One of the principle measures of functional variation in wing shape is the length of the wing relative to the width, known as aspect ratio. In vertebrates, higher aspect ratio (longer, thinner wings) is predicted to give faster and more efficient flight (Norberg, 1989) and has been shown to be associated with migratory species in birds (Mönkkönen, 1995). However, there has been speculation that the benefits of high aspect ratio may be reduced or even reversed at the low Reynolds numbers (a measure of aerodynamic turbulence, with lower numbers corresponding to the viscous forces experienced by small objects) experienced by insects (Ennos, 1989; Wootton, 1992). This speculation, along with the difference in the nature of flight—number, structure and locomotory independence of wings—between birds and insects complicates the formation of hypotheses concerning the implications of variation in flight morphology (Betts & Wootton, 1988; Johansson, Söderquist & Bokma, 2009). The literature on the functional relevance of insect wing morphology is heavily biased towards theory (Dudley, 2002), laboratory studies (Betts & Wootton, 1988; Marden, 1995) and observations of kinematics (Rüppell, 1989; Wakeling & Ellington, 1997a; Wakeling & Ellington, 1997b; Wakeling & Ellington, 1997c) rather than quantitative data collected from the field.

Contrary to predictions for birds, where higher aspect ratios are associated with higher flight speeds (Alerstam et al., 2007), a number of findings point towards lower wing aspect ratio as being beneficial for dispersal in insects. Wing aspect ratio is lower in populations of Pararge aegeria that have recently been founded (Hill, Thomas & Blakeley, 1999). Populations of P. aegeria (Hughes, Dytham & Hill, 2007; Vandewoestijne & Van Dyck, 2011), Drosophila melanogaster (Azevedo et al., 1998), and a number of damselflies (Hassall, Thompson & Harvey, 2009; Taylor & Merriam, 1995) show lower aspect ratio at higher latitudes where temperature reduces the efficiency of flight in ectotherms. This reduction in flight power at lower temperatures has been demonstrated in a number of laboratory systems (Lehmann, 1999) and is likely related to lower wingbeat frequencies at lower temperatures (Dudley, 2002). Since lower wing aspect ratios are associated with greater dispersal ability, it could be that a decline in aspect ratio compensates for this decline in wingbeat frequency (Stalker, 1980). Other studies have shown higher wing aspect ratio only in species of damselflies with expanding range margins (Hassall, Thompson & Harvey, 2008), and those marginal populations exhibit wing shapes that deviate progressively away from the species average closer to the range margin (Hassall & Thompson, 2008). Studies using common garden rearing of Drosophila from a range of latitudes have shown that individuals reared at lower temperatures have lower aspect ratio (Azevedo et al., 1998). While there is no clear relationship between aspect ratio and flight speed in butterflies (Berwaerts, Matthysen & Van Dyck, 2008; but cf Berwaerts, Van Dyck & Aerts, 2002; Dudley, 1990), species in which males “patrol” (i.e., exhibit prolonged flight) tend to have lower aspect ratios (Wickman, 1992). Chironomid females have broader wings (characteristic of lower aspect ratio) to assist with flying for long periods between habitat patches (McLachlan, 1986). While there are exceptions (increased fragmentation does not correlate with aspect ratio in Plebejus argus (Thomas, Hill & Lewis, 1998) or Pararge aegeria (Merckx & Van Dyck, 2006)) these findings seem to suggest that lower wing aspect ratio in insects is associated with greater dispersal.

Odonata have been shown to be sensitive to temperature in a number of life history traits (Hassall & Thompson, 2008) and are responding to climate change by advancing phenology (Hassall et al., 2007) and expanding their ranges poleward (Hassall & Thompson, 2010; Hickling et al., 2006). As a result, odonates would be expected to follow the same geographical patterns as those described above: a decrease in wing aspect ratio to compensate for low wingbeat frequencies at low temperatures (as seen in Diptera), and a further decrease if the species is expanding its range (as seen in Lepidoptera). Wing morphology in Odonata may also be affected by a combination of sexual selection during intrasexual, agonistic interactions, intersexual courtship displays and dispersal (Johansson, Söderquist & Bokma, 2009). In the field, intrasexual territorial contests in Calopteryx maculata are determined by fat reserves (Marden & Rollins, 1994; Marden & Waage, 1990) and contests in Plathemis lydia are determined by flight muscle ratio (Marden, 1989). In both cases, aspect ratio was shown not to influence the outcome of the contests. Sexual selection on courtship displays focuses on patterns of pigmentation in Calopteryx species (Siva-Jothy, 1999; Waage, 1973). However, wing shape has been shown to vary with landscape structure in C. maculata (Taylor & Merriam, 1995) and between some closely-related species of Calopterygidae in Europe (Sadeghi, Adriaens & Dumont, 2009), although not all species exhibited distinct wing shapes. Based on these results, it seems that wing shape variation is under natural selection due to dispersal (within or between sites), rather than sexual selection.

Based on the reasoning presented above, I evaluate the hypothesis that a positive relationship would be found between temperature and aspect ratio to compensate for lower flight efficiency at lower temperatures. Uncertainties over the ecological role of morphology variation may stem from the partial sampling of geographical ranges (Hassall, 2013). Limited sampling of non-linear trends that occur over large spatial scales may produce misleading results and so I provide an analysis of wing shape variation across almost the entire range of the damselfly Calopteryx maculata in North America.

Methods

A total of 907 specimens of male C. maculata were collected from 34 sites across the range by 25 collectors (Fig. 1, Table 1). Collections took place between 13 May and 7 August 2010 and mean sample size from each site varied between 4 and 84 individuals (mean = 26.7 ± 2.9 SE, details of sample sizes and mean measurements can be found in Table 1). Wings were dissected from the body as close to the thorax as possible and mounted on adhesive tape (Scotch Matte Finish Magic Tape). Wings were scanned using the slide scanner on an Epson V500 PHOTO flatbed scanner with fixed exposure at 1200dpi. Wing length (the length from the costal end of the vein separating the arculus from the discoidal cell to the tip of the wing) and wing area were calculated for each of the four wings on each individual. All measurements were carried out in ImageJ (Rasband, 1997–2007). During measurement, any damage to wings was noted and those measurements (length or area) which could not be accurately quantified were excluded. This resulted in the exclusion of 7 fore wing and 9 hind wing lengths, and 28 fore wing and 45 hind wing areas. Aspect ratio was then calculated separately for both fore and hind wings as wingspan2/wing area (see Table 1 for summary statistics and sample sizes). Raw data for measurements can be found in Table S1.

Figure 1 Calopteryx maculata sampling sites.

(A) The geographic distribution of Calopteryx maculata (light shaded area) in relation to the 34 locations at which specimens were collected. (B) Shows the geographical variation in the maximum temperature of the warmest month across the region.

Table 1 Sampling data for Calopteryx maculata.

Sampling site locations, sample sizes and aspect ratios of wings of male Calopteryx maculata. “Measurements” gives the sample size for the total number of measured specimens, “Geo Morph” gives the sample sizes used in the geometric morphometric analysis (Nfore = sample size for fore wings, Nhind = sample size for hind wings).

							Measurements	Geo Morph	
Region	Site	Latitude	Longitude	Date	Fore wing aspect ratio (±SE)	Hind wing aspect ratio (±SE)	N total	N fore	N hind	N fore	N hind	
Ontario	Blakeney Falls	45.268	−76.250	31/05/10	6.845 (±0.044)	6.392 (±0.037)	23	23	23	10	10	
Ontario	Dorset	45.271	−78.960	31/07/10	7.053 (±0.075)	6.564 (±0.069)	7	6	7	6	7	
Ontario	Heber Down	43.941	−78.988	08/06/10	6.845 (±0.034)	6.380 (±0.034)	20	20	20	10	10	
Ontario	Lucknow	43.954	−81.497	28/07/10	7.018 (±0.041)	6.578 (±0.040)	20	20	19	10	10	
Ontario	North Bay	44.947	−79.471	20/06/10–21/06/10	6.811 (±0.019)	6.372 (±0.019)	84	84	84	10	10	
Ontario	Peterborough	44.315	−78.343	15/06/10	6.792 (±0.048)	6.352 (±0.052)	20	20	20	10	10	
Ontario	Ridgetown	42.439	−81.831	11/07/10	6.707 (±0.048)	6.280 (±0.039)	18	18	18	10	10	
Ontario	Sault Ste Marie	46.582	−84.300	24/06/10–26/06/10	6.651 (±0.025)	6.231 (±0.023)	60	60	59	10	10	
Ontario	Serena Gundy Park	43.716	−79.353	15/07/10	6.772 (±0.042)	6.378 (±0.040)	25	25	25	10	10	
Quebec	Dunany	45.758	−74.304	25/06/10	6.925 (±0.036)	6.457 (±0.040)	15	14	15	10	10	
Quebec	Shawinigan	46.514	−72.679	27/06/10	6.857 (±0.032)	6.491 (±0.059)	33	26	25	10	10	
Arkansas	Smithville	36.235	−91.470	22/05/10–07/08/10	6.382 (±0.027)	6.014 (±0.028)	35	35	33	10	10	
Florida	8 Mile Creek	30.483	−87.326	26/06	6.653 (±0.045)	6.278 (±0.039)	20	19	19	10	10	
Georgia	Conyers Monastery	33.584	−84.073	04/08	6.755 (±0.049)	6.331 (±0.045)	11	11	11	10	10	
Georgia	Rome	34.443	−85.150	18/06/10–27/06/10	6.651 (±0.041)	6.221 (±0.036)	20	19	15	10	10	
Illinois	Rockford	42.211	−88.976	17/07/10	6.332 (±0.040)	5.956 (±0.040)	20	20	20	10	10	
Iowa	Gateway Hills Park	42.008	−93.647	24/06/10	6.298 (±0.037)	5.879 (±0.035)	20	20	20	10	10	
Iowa	Odebolt	42.274	−95.129	15/07/10	6.391 (±0.025)	6.040 (±0.024)	73	73	73	10	10	
Kentucky	Fossil Creek	37.773	−84.561	07/06/10	6.757 (±0.046)	6.265 (±0.036)	25	25	25	10	10	
Maryland	Folly Quarter Creek	39.255	−76.927	13/07/10	6.603 (±0.029)	6.247 (±0.031)	33	32	32	10	10	
Michigan	Johnson Creek	42.399	−83.528	19/06/10–26/06/10	6.826 (±0.041)	6.405 (±0.038)	24	23	21	10	10	
Mississippi	Starkville	33.567	−89.041	05/07/10	6.580 (±0.035)	6.190 (±0.031)	26	26	24	10	10	
Missouri	Eleven Point River	36.793	−91.331	05/06/10	6.279 (±0.047)	5.885 (±0.042)	12	12	12	10	10	
Missouri	White River	36.654	−92.230	05/06/10	6.273 (±0.028)	5.903 (±0.028)	25	24	21	10	10	
Nebraska	Chappell	41.083	−102.467	30/06/10	6.408 (±0.065)	6.070 (±0.061)	6	6	6	6	6	
Nebraska	Kimball	41.232	−103.843	01/07/10	6.401 (±0.030)	6.038 (±0.030)	32	32	32	10	10	
Nebraska	Leigh	41.701	−97.247	21/06/10	6.359 (±0.034)	5.963 (±0.034)	25	23	22	10	10	
Ohio	Mt Vernon	40.405	−82.487	16/06/10	6.748 (±0.023)	6.300 (±0.025)	40	39	39	10	10	
South Carolina	Four Holes Swamp	33.212	−80.348	14/07/10	6.782 (±0.059)	6.445 (±0.046)	21	21	21	10	10	
South Carolina	Little Creek	34.842	−82.402	15/07/10	6.777 (±0.040)	6.529 (±0.050)	29	28	28	10	10	
Texas	Powderly	33.753	−95.605	13/05/10	6.287 (±0.033)	5.929 (±0.030)	22	19	18	10	10	
Vermont	Lamoille River	44.681	−73.068	18/06/10	6.873 (±0.123)	6.473 (±0.112)	4	4	4	4	4	
Vermont	West Haven	43.624	−73.362	24/07/10	6.688 (±0.037)	6.277 (±0.035)	17	11	10	10	10	
Vermont	Winooski River	46.352	−72.571	04/07/10–18/07/10	6.895 (±0.034)	6.477 (±0.028)	42	42	41	10	10	

It has been suggested that wing aspect ratio does not provide sufficient detail to be morphologically informative in butterflies (Betts & Wootton, 1988) or dragonflies (Johansson, Söderquist & Bokma, 2009). Therefore, in addition to calculating aspect ratio, I also use geometric morphometrics to derive descriptors of the shape of the wing. A subset of up to 10 individuals from each site were selected at random and a set of 14 landmarks were digitised on 1 fore wing and 1 hind wing (Fig. 2) using tpsDig2 (v.2.12, Rohlf, 2008). Mean locations for each of the 14 landmarks were found for each of the 34 sites. Principal components analysis (PCA) was carried out on these landmarks after Procrustes transformation (to correct for differences in size and rotation of the wing, leaving only shape variation) using the PAST software package (Hammer, Harper & Ryan, 2001). Relationships between the principal components and absolute measurements were investigated using Pearson correlations. Fore and hind wings were compared using paired Hotelling’s t2 tests in PAST to assess whether the two datasets could be combined. Raw data for fore and hind wing geometric morphometric landmarks can be found in Tables S2 and S3, respectively.

Figure 2 Wing landmarks for Calopteryx maculata.

This figure shows the locations of 14 landmarks on the wing of Calopteryx maculata that were digitised and then analysed using geometric morphometrics to describe wing shape.

Bioclim temperature variables (BIO1–BIO11) were extracted for each site from the WORLDCLIM dataset (Haylock et al., 2008) to test the central hypothesis of the study. A large number of candidate variables exist that could be included (11 Bioclim variables, and mean, minimum and maximum temperature for each month). Monthly temperature variables were ignored, as Bioclim variables are more likely to have greater biological relevance. Bioclim variables were subjected to model selection with each of the 11 variables regressed against fore and hind wing aspect ratio and the best-fitting variable selected using Akaike’s information criterion (AIC). Aspect ratio and the informative principal components from the shape analysis were regressed against temperature, latitude, and longitude using linear regressions weighted by the square-root of the sample size. In each case, the models were tested with a quadratic predictor term using AIC to evaluate any improvement in model fit.

Results

Fore and hind wings vary significantly in shape (t2 = 122,500, p ≪ 0.001) and were completely separated along the PC1 axis which explained 80.2% of the variance in shape. As a result, fore and hind wing data are treated separately for the rest of the analysis.

The first three principal components explaining fore and hind wing variation explained 38.7%, 23.2% and 18.6% (total 80.5%) of the variance in fore wing shape and 44.9%, 21.4%, and 12.6% (total 78.9%) of the variance in hind wing shape. PC1 in both cases involved a variation in the width of the wing relative to its length, such that an increase in PC1 leads to a decrease in the width of the wing relative to the length (Fig. 3). The PC2 and PC3 involved more subtle shape changes which were still consistent between wings. PC2 appears to involve a shortening of the pre-nodal region and a blunting of the tip, while PC3 corresponds to a movement of wing area towards the wing tip. PC1 was significantly positively correlated with aspect ratio (fore wings, r = 0.875, p < 0.001; hind wings, r = 0.854, p < 0.001, Fig. 4).

Figure 3 Shape variation in Calopteryx maculata wings.

Deformation plots showing the effect of increasing the value of each principal component on the relative locations of wing landmarks. Arrows indicate the direction and extent of change. Percentages are the percentage of variation explained by each principal component for fore and hind wings, respectively.

Figure 4 Aspect ratio vs. geometric morphometrics.

Relationship between aspect ratio and the first principal component describing variation in wing shape for fore (closed symbols, solid line) and hind wings (open symbols, dotted line) in Calopteryx maculata. Points are mean values from each of 34 sampling sites for both variables.

Aspect ratios for fore and hind wings were very highly correlated (R = 0.978, p < 0.001) and so only statistics for fore wings are presented here. Regression of aspect ratio on latitude showed a substantially improved fit when the quadratic term was included (linear AICc = − 3.4; quadratic AICc = − 10.5; ΔAICc = 7.1). Regression of aspect ratio on longitude showed no improvement in fit when the quadratic term was included (linear AICc = − 35.2; quadratic = − 32.7; ΔAICc = 2.5). The Bioclim temperature variables that best predicted fore wing aspect ratio were Bio5 (maximum temperature of the warmest month, top model) and Bio2 (mean diurnal temperature range, ΔAIC = 1.27). All other variables produced models with ΔAIC > 10 relative to the top model indicating negligible relative explanatory power (Table 2). Bio5 was selected as the temperature variable, as Bio5 models produced greater average support (ΔAIC = 0, ΔAICc = 0.54) than Bio2 (ΔAICc = 0, ΔAICc = 1.27), and represents a measure of absolute temperature (maximum temperature of the warmest month) rather than variability (mean diurnal range), which is closer to the initial hypothesis for the relationship between temperature and aspect ratio. The addition of a quadratic term did not improve the fit of a regression model describing the relationship between aspect ratio and Bio5 (linear AIC = − 18.7; quadratic AIC = − 16.2; ΔAIC = 2.5).

Table 2 Model selection table.

Model fits for linear regression of Bioclim variables (Haylock et al., 2008) on fore and hind wing aspect ratios in Calopteryx maculata.

		Fore wing aspect ratio	Hind wing aspect ratio	
Variable	Definition	logLik	AICc	ΔAICc	logLik	AICc	ΔAICc	
BIO5	Max Temp of Warmest Month	12.743	−18.686	0.000	12.861	−18.923	0.539	
BIO2	Mean Diurnal Range (Mean of monthly (max–min))	12.109	−17.417	1.269	13.131	−19.462	0.000	
BIO10	Mean Temp of Warmest Quarter	7.578	−8.357	10.329	8.860	−10.919	8.542	
BIO3	Isothermality (BIO2/BIO7) (* 100)	5.793	−4.786	13.900	7.327	−7.855	11.607	
BIO1	Annual Mean Temp	4.933	−3.067	15.620	6.790	−6.780	12.682	
BIO11	Mean Temp of Coldest Quarter	3.878	−0.957	17.729	5.973	−5.146	14.315	
BIO8	Mean Temp of Wettest Quarter	3.713	−0.627	18.060	6.395	−5.990	13.472	
BIO6	Min Temp of Coldest Month	3.009	0.782	19.469	5.406	−4.012	15.449	
BIO7	Temper Annual Range (BIO5-BIO6)	2.764	1.271	19.957	6.032	−5.265	14.197	
BIO4	Temp Seasonality (SD *100)	2.354	2.093	20.779	5.056	−3.312	16.150	
BIO9	Mean Temp of Driest Quarter	2.289	2.223	20.909	5.050	−3.301	16.161	

Geographical patterns of wing aspect ratio showed a complex spatial pattern (Fig. 5). There was a U-shaped relationship between aspect ratio and latitude (Table 3; Fig. 6A), explaining 32.5% of the variation in the data. However, this may be due to the distribution of sites (Fig. 5), with most of the mid-latitude sites being found inland in continental areas while northern and southern sites tended to be closer to the coast where temperatures are cooler (Fig. 1B). Indeed, the linear, positive relationship with longitude (Table 3; Fig. 6B), indicating a decline in aspect ratio further west, explained 66.0% of the variation in the data. Aspect ratio was also significantly negatively related to Bio5, which explained 44.6% of the variability in the data (Table 3, Fig. 6C). When the three models were compared, the longitude model explained by far the greatest proportion of the data (Akaike weight ≈ 1; Table 3). However, the geographical distribution of aspect ratio values (Figs. 5 and 6B) suggests that there may be a step-change in wing shape at a certain longitude, rather than a gradual trend.

Figure 5 Aspect ratio variation in Calopteryx maculata.

Distribution of fore wing aspect ratio values for Calopteryx maculata males across the species range (light shaded area) in North America.

Figure 6 Aspect ratio in Calopteryx maculata in relation to latitude, longitude, and temperature.

Relationships between fore wing aspect ratio in Calopteryx maculata and (A) latitude, (B) longitude, and (C) the maximum temperature of the warmest month. Points are mean values from each of 34 sampling sites for both variables.

Table 3 Final models.

Model performance and parameter estimates for regressions of aspect ratio on longitude, latitude, and the maximum temperature of the warmest month.

	Estimate	SE	T	P	R 2	AICc	ΔAICc	
Intercept	8.545	0.236	36.251	<0.001	0.660	−35.2	0.00	
Longitude	0.022	0.003	8.057	<0.001				
Intercept	8.111	0.279	29.106	<0.001	0.446	−18.7	16.53	
Max T warmest month	−0.005	0.001	−5.252	<0.001				
Intercept	14.648	2.771	5.286	<0.001	0.325	−10.5	24.74	
Latitude	−0.434	0.142	−3.048	0.005				
Latitude2	0.006	0.002	3.191	0.003				

Discussion

I provide the first comprehensive assessment of intraspecific variation in wing morphology across almost an entire range in a damselfly. The use of geometric morphometrics to analyse shape confirms that changes in aspect ratio (i.e., changes in the length of the wing relative to the width) constitute the major source of variation between specimens from different sites. I demonstrate a highly significant relationship between temperature (the maximum temperature of the warmest month) and fore wing shape, with higher wing aspect ratios at lower temperatures. The dominant geographical pattern is one of increasing aspect ratio from west to east, which has not been documented in previous studies and may be related to lower maximum temperatures in the western part of the range. A weaker pattern appears to be present with latitude, where there is evidence of higher aspect ratio at the northern and southern range margins.

The literature on the functional relevance of aspect ratio has produced conflicting findings, but the present study offers some insights into this phenomenon that are consistent with previous studies in odonates. The presence of higher aspect ratio wings in regions that experience lower temperatures and at range margins is consistent with previous studies that found higher aspect ratios in cases where flight was more demanding. For example, higher aspect ratios have been associated with populations of calopterygid damselflies inhabiting fragmented habitat (Taylor & Merriam, 1995) and at the expanding edge of the geographical range (Hassall, Thompson & Harvey, 2009). Models predict that improved dispersal should evolve at range margins in response to lower habitat persistence or range expansion (Travis & Dytham, 1999), and these predictions are supported by observations in butterflies (Hill, Thomas & Blakeley, 1999). However, due to the observational nature of this study I cannot disentangle the effects of selection from those of phenotypic plasticity. Indeed, previous studies have demonstrated that while some flight morphological parameters are under genetic control, wing aspect ratio shows a plastic response to the environment in Drosophila (Azevedo et al., 1998). Note that while this study found evidence for a U-shaped relationship between latitude and aspect ratio, the western range margin appears to be associated with very low aspect ratio which is inconsistent with the range margin being associated with high aspect ratio wings. Indeed, the presence of the U-shaped relationship is more likely to be an artefact of the arrangement of sampling sites: the southern sites also tend to be in the eastern part of the range where the aspect ratio is highest (Fig. 5). If it is maximum summer temperature that is driving the variation in wing shape then it might be predicted that there would be little latitudinal pattern in aspects ratio, as maximum summer temperature does not vary consistently with latitude (Fig. 1B). Instead, the temperature variation in the summer tends to be associated with inland vs. coastal areas, with cooler climates in regions closer to the oceans. This coastal buffering of maximum summer temperature, even operating at a scale of 100 s of km (shown in Fig. 1B), provides a potential explanation of the relationship between longitude and wing shape.

It is generally considered that higher aspect ratios provide a benefit for longer-distance flight (Mönkkönen, 1995), efficient, gliding flight (Ennos, 1989), and flight at lower temperatures (Azevedo et al., 1998). A mechanism for this pattern might be provided by Marden’s (1987) observation that wing aspect ratio is negatively related to lift production (controlling for body mass and flight muscle ratio) in conventional wingbeats, but that this is reversed in the case of clap-and-fling wingbeats of the sort used by Calopterygidae. Hence higher aspect ratios generate more lift in Calopteryx sp. which would enhance flight efficiency. However, this is equivocal in Lepidoptera (Betts & Wootton, 1988) where previous studies have found lower aspect ratio at lower temperatures (Vandewoestijne & Van Dyck, 2011). There remains a gap in the literature that needs to be filled with flight laboratory experiments of the functional implications of aspect ratio variation in odonates and other insects as have been carried out in some butterflies (Berwaerts, Matthysen & Van Dyck, 2008; Berwaerts, Van Dyck & Aerts, 2002; Davis et al., 2012). In particular, a test of the hypothesis that higher variation in aspect ratio can enhance flight efficiency at lower temperatures in odonates is warranted given the increasing evidence for the correlation between aspect ratio and temperature.

The association between maximum temperature in the warmest month (which is associated with peaks in emergence in most odonates, Dijkstra & Lewington, 2006) makes sense given the vast quantities of energy expended by insects during this period. Calopteryx males, in particular, compete for and hold territories as well as undertaking extensive aerial contests with competitor males that are energetic wars of attrition (Marden & Waage, 1990; Plaistow & Siva-Jothy, 1996). The small benefit in terms of increased lift from the change in wing shape may benefit males during these activities. However, analysis of these conflicts in Calopteryx virgo showed that there was no difference in aspect ratio between winners and losers (Bots et al., 2012). Given the theoretical benefits and the observed interpopulation variation in aspect ratio, it is surprising that there has not been evolution to a biomechanical optimum across the species. One potential explanation is that aspect ratio is not heritable, but rather is determined by environmental factors as has been shown in Drosophila (Azevedo et al., 1998). It has been proposed that the fore and hind wings of Calopteryx sp. have evolved under natural and sexual selection, respectively (Outomuro, Bokma & Johansson, 2012), but many studies of this kind have failed to sample from a wide geographical range and so the extent to which the findings of those studies can be generalised is unclear.

Previous studies have questioned the use of aspect ratio as a single numerical metric describing wing shape in insects, due to its inability to represent the complexity of wing morphology (Betts & Wootton, 1988; Johansson, Söderquist & Bokma, 2009). However, I find that a complex method of shape analysis using geometric morphometrics yields patterns that strongly resemble variation in the simpler concept of aspect ratio. However, it is clear from the explanatory power of those principal components that correlate with aspect ratio (38.7% and 44.9%) that there is a great deal of variability in addition to this dimension. It is worth noting that insects exhibit a great deal of variation in aspect ratio. Odonates have high aspect ratios compared to some other insects, for example Drosophila virilis with an aspect ratio of 2 (Vogel, 1957), and Bombus terrestris with an aspect ratio of 6.4. However, butterflies show higher aspect ratios of 9.8-10.5 in Pararge aegeria (Berwaerts, Matthysen & Van Dyck, 2008; Berwaerts, Van Dyck & Aerts, 2002). The data presented here show aspect ratios of hind wings between 5.61 and 7.79 and of forewings between 5.70 and 7.56. Aeshna cyanea, a large odonate, exhibits aspect ratio of 8.4 and 11.6 for hind and fore wings, respectively (Ellington, 1984). What makes the odonate wing very different is the extent of the venation in odonate wings compared to other taxa. This venation may be associated with the pleating of the wing, which enhances aerodynamic performance relative to a smooth with of the same shape (Vargas, Mittal & Dong, 2008).

The results presented here demonstrate clear geographical variation in flight morphology in a damselfly across almost its entire range. While the other studies investigating geographical variation in odonate morphology have focused on north–south transects (Johansson, 2003), there are clearly important patterns occurring along the east–west axis of the range highlighting the need to consider range-wide surveys to understand macroecological and macroevolutionary patterns (Hassall, 2013; Hassall, 2014). From the survey of studies that have included aspect ratio, it is clear that laboratory studies are needed to clarify the relationship between form and function in odonate wing shape.

Supplemental Information

Supplemental Information 1 Supplementary information containing measurements and geometric morphometric landmarks for specimens of Calopteryx maculata

Click here for additional data file.

I am extremely grateful to Arne Iserbyt, Mary Burnham, Chris Lewis, Shari Sokay, Darrin OBrien, Fred Sibley, Giff Beaton, George Harp, George Sims, Harris Luckham, John Abbott, Joseph Carson, Jeni Eggers and Eliott Porter, Jeffrey Willers, Michael Blust, Marion Dobbs, Mark Musselman, Pat Heithaus, Rick Abad, Ryan Spafford, Steve Hummel, Sarah Richer, Timothy Sesterhenn, William Lamp and Wade Worthen for giving so graciously of their time to assist with collections. Carley Centen provided valuable assistance in the field and Tom Langen provided assistance with logistics.

Additional Information and Declarations

Competing Interests

Author Contributions

Data Availability

The author declares there are no competing interests.

Christopher Hassall conceived and designed the experiments, performed the experiments, analyzed the data, contributed reagents/materials/analysis tools, wrote the paper, prepared figures and/or tables, reviewed drafts of the paper.

The following information was supplied regarding the deposition of related data:

Figshare: 10.6084/m9.figshare.1468360.

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
