# Peer review of "Strong geographical variation in wing aspect ratio of a damselfly, Calopteryx maculata (Odonata: Zygoptera)"

_PeerJ, doi:10.7717/peerj.1219_

## Round 0.1 · original submission · Minor Revisions

Please address all comments raised by the reviewers, especially the ones concerning experimental design made by Reviewer 2.

·

Basic reporting

No comments

Experimental design

No comments

Validity of the findings

Basic data are provided in Table 1. I do not know if this is enough for PeerJ stantards or if all the data have to be published in a repository.

Additional comments

This paper analyses wing morphology of a calopterygid damselfly along its entire range, and tests hypothesis about the influence of natural selection on wing morphology. It has adequate sample size, well described methods and interesting results. The paper is easily readable and makes a good review of the literature, offering suggestions for future work. My comments are therefore minor.

1. Line 69. This is the first time that the study animal is mentioned, so the generic name should be in full, Calopteryx maculata.
2. Line 123. “The elbow of the scree plot” is difficult to visualize without a figure!
3. Line 137 and following. The use of the word “significant” is tied to a precise meaning in statistics, in the framework of the frequentist approach of null hypothesis testing, and should not be used in the context of Information theory (AIC). Note that the same word is used to refer to “significantly related to temperature” effects in line 142, and in that case it means p<0.05, which is not the case for AIC values<2. Then I would suggest to use another work, like “important”, “substantial”, or similar, when referring to AIC values.
4. Figure 3 is upside-down! (At least compared to Figure 2).

Reviewer 2 ·

Basic reporting

This manuscript looks at the variation in wing morphology, specifically aspect ratio, within the range of a North American damselfly. This is an excellent contribution to the field, the main results being variation in wing morphology longitudinally across this species’ range, driven by the seasonality of precipitation.

I feel that this manuscript could benefit from a better connection between the introduction and the results/discussion. Specifically in the introduction, some topics are glossed over, including 1) why ectotherm flight is expected to be less efficient at lower temperatures (and how this might affect wing aspect ratio), and 2) why we would expect there to be variation in this study system, especially with respect to range expansions in response to climate change (and therefore temperature).

Experimental design

No temperature data are presented, so the reader does not have a good idea how temperature varies across the range of this species. The same is true for seasonality of precipitation. Maps of these data would put the study more in context within the landscape. The author also points out at the end of the manuscript that extremes in temperature and precipitation are likely to be the main driving force of range expansions, but only mean temperature is used. There are many other variables within the Worldclim data which might more accurately represent the aspects of temperature which are influencing odonates, and these should at least be included in addition (or instead of) to mean temperature.

Finally, the author indicates that the samples cover the entire range of this species, yet there are clearly data lacking in the northwest, and especially northeast (Maine and Canadian Maritimes). These additional data would add excellent coverage to this dataset, especially considering these populations are likely to be those at range margins and therefore have aspect ratios deviating from the mean of more centrally located populations. If still lacking these data, the author should change the language to reflect almost complete coverage of this species’ geographic range.

Validity of the findings

No Comments

Additional comments

Introduction:
Line 42: define Reynolds number, or avoid using this term.

Line 43: “this fact” is not a fact, but speculation as defined in the previous sentence. Please reword

Line 45: odonate discussion seems too specific here, keep general in this paragraph, and perhaps mention odonates as an example

Line 48: emphasize not just field observations, but quantitative data collected from the field!

Line 50: provide references for high wing aspect being beneficial in birds

Line 54-55: provide reference for lower temperatures reducing the efficiency of flight for ectotherms.
Please explain this mechanism more clearly.

Line 66: some general introduction of the study species would be good here, especially as it relates to range expansion/flight abilities as this seems to be a driving hypothesis of why the author is looking at wing morphology.

Line 80: the relationship between temperature and flight efficiency has not been thoroughly introduced


Methods:
Line 105: what was the purpose of the procrustes transformation? Procrustes is generally used to compare the outputs of ordinations, not as a transformation. Please explain this better.

Line 110: why use mean annual temp? wouldn’t the extremes, hot or cold, be more influential on C. maculata’s distribution?

Line 155: your sample locations do not cover the entire range of this species, especially lacking in the northeast (maine and the maritimes)

Lines 170-173: You are saying that Taylor and Merriam found high aspect ratios in disturbed landscapes, and that you also found high aspect ratios in your study from sites in the same area north of Ottawa. But then you go on to say that the patterns described in Taylor and Merriam operate at different scales than in this study, but it sounds to me like you have the same results. Both studies found high aspect ratios in the same area. Is this correct? Either way, the purpose of this paragraph is not clear to the reader.

Lines 214 and 215: If extremes are expected to be more important, then why not include some aspects of extreme temperatures (included in the worldclim data as well), instead of mean temperature? You are using seasonality for precipitation, why not something more variable for temperature?


Figure 1: Change the symbology so that the symbols are not overlapping, and the reader has a better idea of where the sample locations are.

---

## Round 0.2 · Minor Revisions

The reviewer has a few comments that still need to be addressed, especially important is the one regarding your conclusion about aspect ratio and temperature.

Reviewer 2 ·

Basic reporting

This revised version of the manuscript is much improved, and the author has done a thorough job in addressing the previous reviewer's comments.

Experimental design

P7, line 162: “As a result, mean annual temperature is used…” This statement is confusing, because you select bioclim variables, but then state that mean annual temperature is used. What is mean annual temperature used for? Please clarify.

Validity of the findings

This is a much improved manuscript, and only one outstanding issue remains for me. If higher aspect ratio is an adaptation for increased flight efficiency in colder climates, then what explains the U shaped distribution with respect to latitude. In other words, why do you find higher aspect ratio at the southern range boundary, instead of only at the northern range boundary? Temperature should not explain this relationship entirely. This could be addressed more thoroughly in the discussion, as I think it is a critical point in understanding the underlying reasons for variation in wing aspect ratio.


Specific Comments:
P 8, line 199: This is the first mention of pigmentation with respect to temperature. Please explain this relationship more.

P10, line 272: If higher aspect ratio is an adaptation for increased flight efficiency in colder climates, then what explains the U shaped distribution with respect to latitude. In other words, why do you find higher aspect ratio at the southern range boundary, instead of only at the northern range boundary?

Page 12, line 409: abrupt transition, topic sentence missing? Please modify

---

## Round 0.3 · accepted · Accept

Thank you for addressing these lingering issues identified by the reviewer.